# Leisure-Time Sedentary Behavior Is Associated with Psychological Distress and Substance Use among School-Going Adolescents in Five Southeast Asian Countries: A Cross-Sectional Study

**DOI:** 10.3390/ijerph16122091

**Published:** 2019-06-13

**Authors:** Supa Pengpid, Karl Peltzer

**Affiliations:** 1ASEAN Institute for Health Development, Mahidol University, Salaya, Nakhon Pathom 73170, Thailand; supaprom@yahoo.com; 2Deputy Vice Chancellor Research and Innovation Office, North West University, Potchefstroom 2531, South Africa

**Keywords:** sedentary behavior, physical activity, psychological distress, tobacco use, alcohol use, adolescents

## Abstract

Sedentary behavior has been found to be associated with poorer mental health. The aim of this study was to estimate associations of sedentary behavior with psychological distress and substance use among adolescents in five Southeast Asian countries. The cross-sectional sample included 32,696 nationally representative samples of school-going adolescents (median age 14 years) from Indonesia, Laos, Philippines, Thailand and Timor-Leste. Leisure-time sedentary behavior, physical activity, psychological distress and substance use were assessed by self-report. Overall, the students engaged in <1 h (35.7%), 1–2 h (31.6%), 3–4 h (18.2%), 5–6 h (7.2%), and 7 or more hours (7.2%) of sedentary time a day. The prevalence of psychological distress was 14.6% single and 8.6% multiple psychological distress, and the prevalence of current tobacco use was 13.9% and current alcohol use 12.5%. In fully adjusted multinomial logistic regression analysis, compared to students who spent less than one hour a day engaged in sedentary leisure time, students who spent three or more hours engaged in leisure-time sedentary behavior were more likely to have single and multiple psychological distress. In fully adjusted logistic regression analysis, five or more hours of leisure-time sedentary behavior was associated with current tobacco use and one or more hours of leisure-time sedentary behavior with current alcohol use. Findings suggest an association of leisure-time sedentary behavior with psychological distress and with substance use in this adolescent population.

## 1. Introduction

During the adolescence period, a significant risk exists for the development of mental and substance use disorders [1], and health risk behaviors are formed which have long lasting effects on health [2,3]. Sedentary behavior (“referring to certain activities in a reclining, seated, or lying position requiring very low energy expenditure” [4]) is one of the health risk behaviors, which may have been significantly increasing in the lives of adolescents [5], and has been implicated with greater risk of obesity, non-communicable diseases and mortality [5,6]. Sedentary behavior may have been increasing because of a change from physically demanding tasks to more knowledge-based activities [7]. This may be reflected by activities such as social media use or computer “chatting”, being associated with poorer mental health, such as depression [7,8].

Growing evidence seems to show links between sedentary behavior and poor mental health among adolescents [2,9,10,11,12]. In a systematic review among adolescents, Hoare et al. [2] found a strong relationship between leisure screen time and both depressive symptoms and psychological distress; poorer mental health status was found in those with 2–3 h or more per day of screen time [2]. In a more recent review, Rodriguez-Ayllon et al. [13] found that greater amounts of sedentary behavior was associated with poorer mental health (depression) but associations with psychological distress were unclear. Using data from the multi-country “Global School-Based Student Health Survey (GSHS)”, Vancamfort et al. [9,10,11,12] found that leisure-time sedentary behavior was associated with various poor mental health indicators (depressive symptoms, loneliness, suicide attempt and anxiety-induced sleep disturbance). 

Less evidence was found for an association between sedentary behavior and substance use among adolescents [14,15,16]. Among school-going adolescents in Slovenia, computer use time (≥2 h/day) was associated with tobacco and alcohol use [14], among Hispanic adolescents, sedentary behavior was associated with alcohol use [15], and in a study among in-school adolescents in eight African countries, leisure-time sedentary behavior (3–4 h or more/day) was associated with tobacco, alcohol and drug use [16]. We are not aware of studies investigating the relationship between sedentary behavior and psychological distress and substance use among adolescents in Asia.

More research is needed to study the independent associations of sedentary behavior with psychological distress and substance use among adolescents, in particular in Southeast Asia. In a study among adolescents in seven Southeast Asian countries, an association between loneliness, current alcohol use and sedentary behavior was found [17]. Possible mechanisms for negative effects of sedentary behavior on mental health are not clear [13]. “It is possible that the beneficial pathophysiological, social and general health benefits of being active may be omitted when sedentary, which may have a negative impact on mental health. Adolescents who experience poorer mental health may lack of motivation to be physically active and may turn to screen based activities requiring little effort as a coping mechanism, and therefore lose such protective effects of physical activity.” [2]. It was hypothesized that higher leisure-time sedentary behavior will be associated with a higher prevalence of psychological distress and substance use.

This study aimed to investigate the associations of leisure-time sedentary behavior with psychological distress and with substance use among school-going adolescents in five Southeast Asian countries.

## 2. Materials and Methods 

### 2.1. Sample and Procedure

Cross-sectional data from the GSHS of five Southeast Asian countries (which had collected data as recent as 2015) were analyzed. The GSHS uses a “cluster sampling design in two stages (schools and classrooms) in order to produce nationally representative samples of school children in middle schools” [18]. “Students completed a self-administered questionnaire under the supervision of trained survey administrators” [18]. Country level ethics review boards approved the GSHS, and “informed consent was obtained from the students, parents and/or school officials” [18].

### 2.2. Measures

The questionnaire used was from the GSHS [18] and is shown in Table 1. The psychological distress items (no close friends, loneliness, anxiety, suicidal ideation and suicide attempt) were summed, and grouped into 0 = 0 no, 1 = 1 single and 2–5 = 2 multiple. Underweight was defined “as less than −2SD from median for BMI by age and sex”, and overweight or obesity was defined as “more than +1 standard deviation (SD) from the median body mass index by age and sex” [19]. Adequate fruit consumption was classified as “two or more servings in a day and adequate consumption of vegetables as three or more servings a day” [20]. Adequate physical activity was defined as “at least 60 min of moderate to vigorous-intensity physical activity daily” [21]. The four items on parental or guardian support were summed, and classified into three groups, 0–1 low, 2 medium and 3–4 high support. Variables for inclusion as covariates were selected based on a previous literature review [2].

### 2.3. Data Analysis

Data analysis was conducted with STATA software version 15.0 (Stata Corporation, College Station, TX, USA), taking into account the complex study design. Multinomial logistic regression was used to assess the associations between leisure-time sedentary behavior and single and multiple psychological distress (with no psychological distress as reference category), in Model 1: adjusted for country, and in Model 2: adjusted for country, age, sex, experience of hunger, physical activity, body weight status, fruit and vegetable consumption, soft drink consumption, fast food consumption, bullying victimization, peer support and parental support. Multivariate logistic regression was used to assess the associations of leisure-time sedentary behavior with the prevalence of current tobacco use and current alcohol use, in Model 1: adjusted for country, and in Model 2: adjusted for country, age, sex, experience of hunger, physical activity, body weight status, fruit and vegetable consumption, soft drink consumption, fast food consumption, bullying victimization, peer support and parental support. A *p*-value of <0.05 was considered significant.

## 3. Results

### 3.1. Sample Characteristics

The total sample 32,696 school-going adolescents from five Southeast Asian countries, median age 14 years (interquartile range = 2 years), with complete sedentary behavior measurements from 10,922 students from Indonesia (overall response rate = 94%), 3671 from Laos (72%), 8665 from the Philippines (79%), 5780 from Thailand (89%) and 3658 from Timor-Leste (response rate = 79%) [18]. Country participation ranged from Laos and Timor-Leste 11.2% to Indonesia 33.4%. Overall, the students engaged in <1 h (35.7%), 1–2 h (31.6%), 3–4 h (18.2%), 5–6 h (7.2%), and 7 or more hours (7.2%) of leisure-time sedentary behavior a day. The prevalence of psychological distress was 14.6% single and 8.6% multiple psychological distress, and the prevalence of current tobacco use was 13.9% and current alcohol use 12.5% (see Table 2).

### 3.2. Associations between Leisure-Time Sedentary Behaviour and Psychological Distress

In fully adjusted multinomial logistic regression analysis, compared to students who spent less than one hour a day engaged in leisure-time sedentary behavior, students who spent three or more hours engaged in leisure-time sedentary behavior were more likely to have single and multiple psychological distress (see Table 3).

### 3.3. Associations of Leisure-Time Sedentary Behaviour with Tobacco and Alcohol Use

In fully adjusted logistic regression analysis, five or more hours of leisure-time sedentary behavior was associated with current tobacco use, and 1–2 h or more of sedentary time was associated with current alcohol use (see Table 4).

## 4. Discussion

This study aimed to estimate associations of leisure-time sedentary behavior with psychological distress and with substance use among adolescents in Southeast Asia. Consistent with a previous review and multi-country studies among adolescents [2,9,10,11,12], this study found that higher leisure-time sedentary behavior (≥3 h) increased the risk ratio for psychological distress, after adjusting for relevant confounders, in this population of school-going adolescents. Further, this study found that compared to <1 h leisure-time sedentary behavior, students with five or more hours leisure time sedentary behavior had a higher odds of current tobacco use and students with one or more hours leisure-time sedentary behavior had a higher odds of current alcohol use. This finding is in agreement with studies in USA and Africa [14,16].

The association between leisure-time sedentary behavior and psychological distress was stronger in students with multiple psychological distress (Adjusted Relative Risk Ratio (ARRR): 2.39, Confidence Interval (CI): 1.86, 3.09), compared to students with single psychological distress (ARRR: 1.49, CI: 1.19, 1.88). This study also found a dose-dependent association between leisure-time sedentary behavior and having multiple psychological distress, current tobacco and alcohol use, as found previously with poor mental health outcomes [7,8]. The prevalence of multiple psychological distress increased from 6.9% with <1 h leisure-time sedentary behavior per day to 15.4% with seven or more hours leisure-time sedentary behavior per day. Similarly, current tobacco use increased from 11.9% to 18.0% and current alcohol use increased from 9.7% with <1 h leisure-time sedentary behavior per day to 20.9% with seven or more hours leisure-time sedentary behavior per day. Some experimental research seems to suggest a causal relationship from sedentary behavior to psychological distress [22]. Moreover, adolescents who have psychological distress may be more likely to develop more sedentary behavior [2,7]. In addition, increased social media use may be by itself linked to increased psychological distress [8,23,24].

The association between sedentary behavior and substance use was stronger for alcohol than for tobacco use. Chiao et al. [25] found in a longitudinal study that adolescent internet use was associated with alcohol use and cigarette smoking. It is possible that increased and possibly problematic social media use as a behavioral addiction develops in co-occurrence with substance use (tobacco and alcohol) addiction [14,26]. 

### Study Limitations

Based on the cross-sectional nature of this study, no causal conclusions can be drawn. Further, the leisure-time sedentary behavior and other variables were assessed by a self-reported questionnaire. Future studies should include both self-reported and objective measures of sedentary behavior. Another limitation was that screen time was only assessed together with overall leisure-time sedentary behavior, and we cannot therefore discern effects of screen time and other sedentary behavior.

## 5. Conclusions

Our findings confirm previous results demonstrating an association between leisure-time sedentary behavior and psychological distress. In addition, findings from a few previous studies are extended by showing associations of higher leisure-time sedentary behavior with tobacco and alcohol use. Mental health and substance use intervention strategies among school-going adolescents may include reducing and/or interrupting sedentary behavior.

## Figures and Tables

**Table 1 ijerph-16-02091-t001:** Variable description.

Variables	Question	Response Options
Age	“How old are you?”	“11 years old or younger to 18 years old or older”
Sex	“What is your sex?”	“Male, Female”
Hunger	“During the past 30 days, how often did you go hungry because there was not enough food in your home?"	“1 = never to 5 = always (coded 1–3 = 0 and 4–5 = 1)”
Leisure-time sedentary behavior	“How much time do you spend during a typical or usual day sitting and watching television, playing computer games, talking with friends, or doing other sitting activities, such as …country specific examples?”	“1 = less than 1 h per day; 2 = 1–2 h/day; 3 = 3–4 h/day; 4 = 5–6 h/day; 5 = 7–8 h/day and 6 = 8 or more hours per day”
Physical activity	“During the past 7 days, on how many days were you physically active for a total of at least 60 min per day? ADD UP ALL THE TIME YOU SPENT IN ANY KIND OF PHYSICAL ACTIVITY EACH DAY”	“1 = 0 days to 8 = 7 days (coded 1–7 = 0 and 8 = 1)”
Height	“How tall are you without your shoes on?”	cm
Body weight	“How much do you weigh without your shoes on?”	kg
Fruits	“During the past 30 days, how many times per day did you usually eat fruit such as… country specific names?”	“1 = I did not eat fruit during the past 30 days to 7 = 5 or more times per day (coded 1–3 = 0 and 4–8 = 1)”
Vegetables	“During the past 30 days, how many times per day did you usually eat vegetables, such as …country specific names?”	“I did not eat vegetables during the past 30 days to 7 = 5 or more times per day (coded 1–4 = 0 and 5–8 = 1”
Soft drinks	“During the past 30 days, how many times per day did you usually drink carbonated soft drinks, such as… country specific names?”	“1 = not in the past days to 7 = 5 or more times per day (coded 1 = 0 and 2–7 = 1)”
Fast food	“During the past seven days, on how many days did you eat food from a fast food restaurant, such as… country specific names?”	“1 = 0 to 8 = 7 days (coded 1–3 = 0 and 4–8 = 1)”
Bullied	“During the past 30 days, on how many days were you bullied?”	“1 = 0 days to 7 = All 30 days (coded 1 = 0 and 2–7 = 1)”
Peer support	“During the past 30 days, how often were most of the students in your school kind and helpful?”	“1 = never to 5 = always (coded 1–3 = 0 and 4–5 = 1)”
Parental/guardian supervision	“During the past 30 days, how often did your parents or guardians check to see if your homework was done?”	“1 = never to 5 = always (coded 1–3 = 0 and 4–5 = 1)”
Parental/guardian connectedness	“During the past 30 days, how often did your parents or guardians understand your problems and worries?”	“1 = never to 5 = always (coded 1–3 = 0 and 4–5 = 1)”
Parental/guardian bonding	“During the past 30 days, how often did your parents or guardians really know what you were doing with your free time?”	“1 = never to 5 = always (coded 1–3 = 0 and 4–5 = 1)”
Parental/guardian respect for privacy	“During the past 30 days, how often did your parents or guardians go through your things without your approval?”	“1 = never to 5 = always (coded 1–3 = 0 and 4–5 = 1)”
	**Psychological distress indicators**	
No close friends	“How many close friends do you have?”	“1 = 0 to 4 = 3 or more (coded 1+ = 0, 0 = 1)”
Loneliness	“During the past 12 months, how often have you felt lonely?”	“1 = never to 5 = always (coded 1–3 = 0 and 4–5 = 1)”
Anxiety	“During the past 12 months, how often have you been so worried about something that you could not sleep at night?”	“1 = never to 5 = always (coded 1–3 = 0 and 4–5 = 1)”
Suicide ideation	“During the past 12 months, did you ever seriously consider attempting suicide?”	“Yes, No”
Suicide attempt	“During the past 12 months, how many times did you actually attempt suicide?”	“1 = 0 times to 5 = 6 or more times (coded 1 = 0 and 2–5 = 1)”
	**Substance use**	
Past month or current tobacco use	“During the past 30 days, on how many days did you smoke cigarettes/use any tobacco products other than cigarettes, such as such as country examples…?”	“1 = 0 days to 7 = All 30 days (coded 1 = 0 and 2–7 = 1)”
Current alcohol use	“During the past 30 days, on how many days did you have at least one drink containing alcohol?”	“1 = 0 days to 7 = All 30 days (coded 1 = 0 and 2–7 = 1)”

**Table 2 ijerph-16-02091-t002:** Sample characteristics by sedentary behavior, psychological distress and substance use.

Variable		Leisure-Time Sedentary Behavior (in hours/day)					Psychological Distress		Tobacco Use	Alcohol Use
	Sample	<1	1–2	3–4	5–6	7 or More	Single	Multiple	Past Month	Past Month
	*N* (%)	%	%	%	%	%	%	%	%	%
All	32,696	35.7	31.6	18.2	7.2	7.2	14.6	8.6	13.9	12.5
Country										
Indonesia	10,922 (33.4)	37.1	35.6	16.8	5.4	5.2	10.9	4.5	12.8	4.4
Laos	3671 (11.2)	41.9	37.0	13.7	3.3	4.1	13.5	3.4	6.2	29.3
Philippines	8665 (26.5)	40.8	28.0	17.2	6.7	7.3	21.0	14.5	15.8	21.2
Thailand	5780 (17.7)	19.1	24.6	26.0	15.7	14.6	14.4	12.0	13.9	22.0
Timor-Leste	3658 (11.2)	50.7	34.5	7.6	2.2	4.9	23.4	11.0	30.2	17.0
Age in years										
13 or less	9420 (28.9)	42.1	32.8	14.9	5.1	5.2	12.8	6.3	10.0	6.7
14	6732 (20.7)	34.4	33.3	17.8	7.0	7.5	14.5	8.5	13.3	10.2
15	6313 (19.4)	34.5	30.7	19.3	7.7	7.8	16.1	10.6	16.1	14.5
16 or more	10,102 (31.0)	28.2	28.9	22.9	10.5	9.6	16.1	10.7	18.7	22.0
Sex										
Female	17,751 (54.8)	35.1	31.8	18.5	7.1	7.5	14.7	9.8	5.5	9.1
Male	14,662 (45.2)	36.4	31.5	17.8	7.4	6.9	14.5	7.4	22.7	16.0
Hunger (mostly/always)	1884 (5.1)	41.8	25.2	15.1	7.5	10.4	25.1	18.1	19.8	17.1
Physical activity (7 days)	3501 (10.7)	27.0	34.0	19.7	8.8	10.5	13.5	8.2	12.1	10.9
Body weight										
Normal	23,206 (77.2)	35.8	31.5	18.0	7.3	7.4	14.1	8.5	12.6	12.0
Underweight	2796 (8.7)	38.0	31.2	17.2	7.6	6.1	12.6	7.5	14.4	12.0
Overweight/obese	3714 (14.0)	30.7	32.0	21.4	7.9	7.9	13.0	7.2	11.1	9.1
Fruit consumption (≥2/day)	10,690 (36.4)	35.9	32.0	18.1	7.1	6.9	13.8	8.2	12.7	12.0
Vegetable consumption (≥3/day)	8523 (28.3)	38.1	30.4	17.8	6.5	7.3	14.2	8.4	12.2	11.1
Soft drink consumption (≥2/day)	5655 (15.0)	28.9	28.3	21.1	9.8	11.9	17.7	14.0	19.4	21.1
Fast food (≥1/week)	18,205 (56.5)	31.6	31.5	20.0	8.4	8.5	14.9	9.2	14.8	14.2
Bullied	11,078 (36.8)	35.2	30.2	18.3	7.9	8.4	21.6	16.7	21.0	19.7
Peer support (mostly/always)	11,031 (36.9)	33.0	32.4	18.9	7.6	8.1	12.4	7.3	10.5	10.0
Parental/guardian support index										
0	17,003 (51.5)	36.5	32.1	17.5	7.0	7.0	16.6	10.9	18.0	15.7
1	8423 (27.0)	34.1	31.8	19.8	7.5	6.8	12.9	6.4	10.4	9.9
2–3	6281 (21.5)	36.8	31.0	17.7	7.0	7.5	11.8	5.7	7.0	6.9
Psychological distress										
none	23,548 (76.8)	78.7	79.6	74.6	70.5	66.5				
single	4969 (14.6)	14.4	13.0	15.7	16.2	18.1				
multiple	2802 (8.6)	6.9	7.4	9.7	13.3	15.4				
Current tobacco use	4488 (13.9)	11.9	13.8	14.6	18.4	18.0				
Current alcohol use	4801 (12.5)	9.7	11.3	14.4	18.2	20.9				

**Table 3 ijerph-16-02091-t003:** Associations of leisure-time sedentary behavior with single and multiple psychological distress (with no psychological distress as reference category).

Variable	Single Psychological Distress	Multiple Psychological Distress
	ARRR ^1^	ARRR ^1^
Leisure-time sedentary behavior, hours/day		
<1	1 (Reference)	1 (Reference)
1–2	0.95 (0.85, 1.07)	1.16 (0.99, 1.37)
3–4	1.18 (1.04, 1.35) *	1.46 (1.25, 1.71) ***
5–6	1.26 (1.05, 1.35) *	1.99 (1.63, 2.44) ***
7 or more	1.49 (1.22, 1.81) ***	2.42 (1.99, 2.96) ***
	ARRR ^2^	ARRR ^2^
Leisure-time sedentary behavior, hours/day		
<1	1 (Reference)	1 (Reference)
1–2	0.92 (0.81, 1.05)	1.17 (0.96, 1.44)
3–4	1.18 (1.02, 1.36) *	1.50 (1.23, 1.83) ***
5–6	1.29 (1.04, 1.60) *	2.06 (1.56, 2.71) ***
7 or more	1.49 (1.19, 1.88) ***	2.39 (1.86, 3.09) ***

ARRR = Adjusted Relative Risk Ratio; CI = Confidence Interval; ^1^ Adjusted for country, ^2^ Adusted for country, age, sex, experience of hunger, physical activity, body weight status, fruit and vegetable consumption, soft drink consumption, fast food consumption, bullying victimization, peer support and parental support. *** *p* < 0.001, ** *p* < 0.01, * *p* < 0.05.

**Table 4 ijerph-16-02091-t004:** Associations between leisure-time sedentary behavior and tobacco and alcohol use.

Variable	Current Tobacco Use	
	AOR ^1^	AOR ^2^
Leisure-time sedentary behavior, hours/day		
<1	1 (Reference)	1 (Reference)
1–2	1.21 (1.03, 1.43) *	1.10 (0.93, 1.29)
3–4	1.30 (1.10, 1.53) **	1.10 (0.90, 1.34)
5–6	1.71 (1.41, 2.08) ***	1.55 (1.22, 1.98) ***
7 or more	1.66 (1.32, 2.07) ***	1.55 (1.14, 2.11) **
	AOR ^1^	AOR ^2^
Leisure-time sedentary behavior, hours/day		
<1	1 (Reference)	1 (Reference)
1–2	1.32 (1,14, 1,52) ***	1.18 (1.03, 1.35) *
3–4	1.50 (1.31, 1.72) ***	1.31 (1.12, 1.52) ***
5–6	1.76 (1.50, 2.07) ***	1.52 (1.25, 1.85) ***
7 or more	2.06 (1.77, 2.41) ***	1.87 (1.61, 2.18) ***

AOR = Adjusted Odds Ratio; CI = Confidence Interval; ^1^ Adjusted for country, ^2^ Adusted for country, age, sex, experience of hunger, physical activity, body weight status, fruit and vegetable consumption, soft drink consumption, fast food consumption, bullying victimization, peer support and parental support. *** *p* < 0.001, ** *p* < 0.01, * *p* < 0.05.

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
