# Peer review of "Leisure-Time Sedentary Behavior Is Associated with Psychological Distress and Substance Use among School-Going Adolescents in Five Southeast Asian Countries: A Cross-Sectional Study"

_ijerph, 2019, doi:10.3390/ijerph16122091_

Round 1
Reviewer 1 Report
Abstract
Please include one background sentence explaining the rationale for the study.
Presumably, psychological distress and substance use were also assessed via self-report. Please include these details.
Results: It would be more useful to just indicate that the majority of students spent less than 2 hours in sedentary time here.
Results: At present, the findings for psychological distress are confusing – suggest re-wording to simplify or providing % who fell into the “high” range.
Greater consideration of the findings for each outcome (psych distress, alcohol use, tobacco use) is warranted – 5hrs of sedentary time was associated with tobacco use, compared to just 1hr for alcohol use.
Introduction
Please define what you mean by sedentary behaviour in the first paragraph i.e. sitting time? Screen time (what screens?), leisure time only?
The authors should refer to existing literature and theory about why sedentary time is associated with mental health and substance use outcomes.
Suggest explaining why sedentary behaviour is increasing in prevalence and possible links to increasing use of screen time and social media use
The authors should include any relevant research conducted in the five ASEAN countries on this topic.
Line 46,47 – sedentary behaviour (play time), why is play time considered sedentary behaviour?
Methods
Sample - Include total number of students, by country
How long was the student questionnaire?
Did the Global School-based Student Health Survey use validated scales within the questionnaire? For example, has the measure of psychological distress been validated among youth?
Additional measures, such as fruit and vegetable consumption and BMI, are introduced in the “Measures” section. Please introduce these in the background or remove from the main text and leave in the table only.
Table 1 should go in an online appendix, with just the key variables included with the main manuscript.
Results
In the methods, the authors state that psychological distress indicators were summed into low, medium, high – please use this terminology in the analysis and results sections for consistency (instead of one indicators vs multiple indicators).
At present, the findings for psychological distress are confusing – suggest re-wording to simplify or providing % who fell into the “high” range.
Discussion
The discussion is very brief. Greater discussion of the findings, especially their implications, are needed.
General comments
It would be better to have consistent terminology when talking of ‘Leisure-time sedentary behaviour’. Throughout the manuscript this is referred to in different ways including ‘leisure sedentary time’, ‘leisure screen time’, ‘sedentary behaviour’.
Suggest adding ‘cross-sectional study’ to the title
Author Response
Abstract
Please include one background sentence explaining the rationale for the study.
Response: added
Presumably, psychological distress and substance use were also assessed via self-report. Please include these details.
Response: added
Results: It would be more useful to just indicate that the majority of students spent less than 2 hours in sedentary time here.
Response: disagree
Results: At present, the findings for psychological distress are confusing – suggest re-wording to simplify or providing % who fell into the “high” range.
Response: is reworded
Greater consideration of the findings for each outcome (psych distress, alcohol use, tobacco use) is warranted – 5hrs of sedentary time was associated with tobacco use, compared to just 1hr for alcohol use.
Response: each outcome is already reported, as suggested
Introduction
Please define what you mean by sedentary behaviour in the first paragraph i.e. sitting time? Screen time (what screens?), leisure time only?
Response: below is added
Sedentary behaviour refers to certain activities in a reclining, seated, or lying position requiring very low energy expenditure [4]
The authors should refer to existing literature and theory about why sedentary time is associated with mental health and substance use outcomes.
Response: This is explained in the discussion
Suggest explaining why sedentary behaviour is increasing in prevalence and possible links to increasing use of screen time and social media use
Response: below is added
Sedentary behaviour may be increasing because of a change from physically demanding tasks to more knowledge-based activities [6]. This may be reflected by activities such as social media use or computer “chatting”, being associated with poorer mental health, such as depression [6,7].
The authors should include any relevant research conducted in the five ASEAN countries on this topic.
Response: some is added
Line 46,47 – sedentary behaviour (play time), why is play time considered sedentary behaviour?
Response: This study is removed
Methods
Sample - Include total number of students, by country
Response: added
How long was the student questionnaire?
Response: 12 pages
Did the Global School-based Student Health Survey use validated scales within the questionnaire? For example, has the measure of psychological distress been validated among youth?
Response: No
Additional measures, such as fruit and vegetable consumption and BMI, are introduced in the “Measures” section. Please introduce these in the background or remove from the main text and leave in the table only.
Response: these are not additional measures, they are in the Table 1, only classifications are described. Further:
Variables for inclusion as covariates were selected based on a previous literature review [2].
Table 1 should go in an online appendix, with just the key variables included with the main manuscript.
Response: This may be possible, but if doing so, one would have to repeat the lengthy descriptions of the key variables in the text, which would almost double the text for the measures
Results
In the methods, the authors state that psychological distress indicators were summed into low, medium, high – please use this terminology in the analysis and results sections for consistency (instead of one indicators vs multiple indicators).
Response: Under measure, changed to below, and harmonized in text
The psychological distress items (no close friends, loneliness, anxiety, suicidal ideation and suicide attempt) were summed, and grouped into 0=0 no, 1=1 single and 2-5=2 multiple psychological distress.
At present, the findings for psychological distress are confusing – suggest re-wording to simplify or providing % who fell into the “high” range.
Response: corrected as above
Discussion
The discussion is very brief. Greater discussion of the findings, especially their implications, are needed.
Response: This is a brief report. Major issues are discussed and possible implications are described in the conclusion
General comments
It would be better to have consistent terminology when talking of ‘Leisure-time sedentary behaviour’. Throughout the manuscript this is referred to in different ways including ‘leisure sedentary time’, ‘leisure screen time’, ‘sedentary behaviour’.
Response: Corrected accordingly
Suggest adding ‘cross-sectional study’ to the title
R: added
Reviewer 2 Report
Review of the manuscript "Leisure-Time Sedentary Behaviour is Associated with Psychological Distress and Substance Use among School-Going Adolescents in Five ASEAN Countries".
Summary
It is necessary for the authors to summarize in the format "background, methods, results, discussion and conclusions".
Keywords
Country names cannot be keywords.
Please change the word "use" to "consumption".
Introduction
The introduction is very poor and very short. The authors need to improve this section substantially. They must add references and describe the study hypothesis.
Table 1, 3 and 4 should be smaller in size in order to facilitate reading.
It would be advisable for authors to add a section on research limitations.
References
Check the Vancouver regulations, as there are some references that are not adapted.
Add more references, as there are only 22 references.
Author Response
Summary
It is necessary for the authors to summarize in the format "background, methods, results, discussion and conclusions".
Response: Background is added, not structured abstract
Keywords
Country names cannot be keywords.
Response: removed
Please change the word "use" to "consumption".
Response: Disagree
Introduction
The introduction is very poor and very short. The authors need to improve this section substantially. They must add references and describe the study hypothesis.
Response: Some additions are made, including the study hypothesis. As a brief report the introduction be short
Table 1, 3 and 4 should be smaller in size in order to facilitate reading.
Response: Table 1 is reduced with a smaller size, Tables 3 and 4 should be OK
It would be advisable for authors to add a section on research limitations.
Response: this is already there, as below
4.1. Study limitations
Based on the cross-sectional nature of the study, no causal conclusions can be drawn. Further, the leisure-time sedentary behaviour and other variables were assessed by a self-reported questionnaire. Future studies should include both self-reported and objective measures of sedentary behaviour. Another limitation was that screen time was only assessed together with overall leisure-time sedentary behaviour, and we cannot therefore discern effects of screen time and other sedentary behaviour.
References
Check the Vancouver regulations, as there are some references that are not adapted.
Response: Corrected
Add more references, as there are only 22 references.
Response: some more are added, but this is a brief report
Reviewer 3 Report
This manuscript has solid quality in many areas and conveys findings about an important aresa However a few things will make it strong er and more useful
The authors need to carefully articulate why this work is new. They cite many articles, some of which seeminingly covering the same topic The authors must make an argument about what novel and different approach and analyses this articles brings to the field by naming the gap in the literature this this article fills
Why were all the variables dichotomized? They could have been handled differently What was the rationale?
Move Table 1 to supplementary material in the ms.
There are too many quotes in Table 3. Try to describe without quotes.
Author Response
This manuscript has solid quality in many areas and conveys findings about an important aresa However a few things will make it strong er and more useful
The authors need to carefully articulate why this work is new. They cite many articles, some of which seeminingly covering the same topic The authors must make an argument about what novel and different approach and analyses this articles brings to the field by naming the gap in the literature this this article fills
Response: below is added
We are not aware of studies investigating the relationship between sedentary behaviour and psychological distress and substance use in Asia
Why were all the variables dichotomized? They could have been handled differently What was the rationale?
Response: Not all variables are dichotomized. Variables, which were meaningful to combine, were combined, such as psychological distress and parental support, others use recommended cut points such as BMI, physical inactivity, current alcohol use, others are single indices with meaningful out points commonly used, such as past-month bullying victimization.
Move Table 1 to supplementary material in the ms.
Response: This may be possible, but if doing so, one would have to repeat the lengthy descriptions of the key variables in the text, which would almost double the text for the measures
There are too many quotes in Table 3. Try to describe without quotes.
Response: The exact wording of the questions should be in quotes
Round 2
Reviewer 2 Report
Review of the manuscript entitled "Leisure-Time Sedentary Behaviour is Associated with 2 Psychological Distress and Substance Use among 3 School-Going Adolescents in Five ASEAN Countries: 4 A Cross-sectional Study".
The manuscript presented presents a poor review of the literature. 24 articles are too few for introduction and discussion.
Authors are recommended to extend the introductory sections (detailing the variables and hypothesis), and the discussion section with updated studies.
Also in the tables there are titles that are cut, so it is recommended to put a smaller size in the letter, for example in table 2.
Author Response
The manuscript presented presents a poor review of the literature. 24 articles are too few for introduction and discussion.
Response: More is added
Authors are recommended to extend the introductory sections (detailing the variables and hypothesis), and the discussion section with updated studies.
Response: More is added
Also in the tables there are titles that are cut, so it is recommended to put a smaller size in the letter, for example in table 2.
Response: Corrected
Reviewer 3 Report
The authors have made many changes and edits since the first version. The manuscript is much improved
Author Response
(x) English language and style are fine/minor spell check required
Response: Corrected